

# Combined detection of serum IL-6 and CEA contributes to the diagnosis of lung adenocarcinoma *in situ*

Jing Pan[1,2], Wanzhen Zhuang[2,3], Yu Xia[3,4], Zhixin Huang[3,4], Yue Zheng[2,3], Xin Wang[1] and Yi Huang[2,3,5,6]

[1] Department of Clinical Laboratory, Rehabilitation Hospital Affiliated to Fujian University of Traditional Chinese Medicine, Fuzhou, Fujian, China
[2] Shengli Clinical Medical College, Fujian Medical University, Fuzhou, China
[3] Department of Clinical Laboratory, Fujian Provincial Hospital, Fuzhou, China
[4] Integrated Chinese and Western Medicine College, Fujian University of Traditional Chinese Medicine, Fuzhou, Fujian, China
[5] Center for Experimental Research in Clinical Medicine, Fujian Provincial Hospital, Fuzhou, Fujian, China
[6] Fujian Provincial Key Laboratory of Critical Care Medicine, Fujian Provincial Key Laboratory of Cardiovascular Disease, Fuzhou, China

Corresponding authors
Xin Wang, 511844551@qq.com
Yi Huang, huangyi@fjsl.com.cn

## ABSTRACT

**Background**. Effective discrimination of lung adenocarcinoma (LUAD) *in situ* (AIS) from benign pulmonary nodules (BPN) is critical for the early diagnosis of AIS. Our pilot study in a small cohort of 90 serum samples has shown that serum interleukin 6 (IL-6) detection can distinguish AIS from BPN and health controls (HC). In this study, we intend to comprehensively define the diagnostic value of individual and combined detection of serum IL-6 related to the traditional tumor markers carcinoembryonic antigen (CEA) and cytokeratin 19 fragment (CYFRA21-1) for AIS.

**Methods**. The diagnostic performance of serum IL-6 along with CEA and CYFRA21-1 were evaluated in a large cohort of 300 serum samples by a chemiluminescence immunoassay and an electrochemiluminescence immunoassay. A training set comprised of 65 AIS, 65 BPN, and 65 HC samples was used to develop the predictive model for AIS. Data obtained from an independent validation set was applied to evaluate and validate the predictive model.

**Results**. In the training set, the levels of serum IL-6 and CEA in the AIS group were significantly higher than those in the BPN/HC group ($P < 0.05$). There was no significant difference in serum CYFRA21-1 levels between the AIS group and the BPN/HC group ($P > 0.05$). Serum IL-6 and CEA levels for AIS patients showed an area under the curve (AUC) of 0.622 with 23.1% sensitivity at 90.7% specificity, and an AUC of 0.672 with 24.6% sensitivity at 97.6% specificity, respectively. The combination of serum IL-6 and CEA presented an AUC of 0.739, with 60.0% sensitivity at 95.4% specificity. The combination of serum IL-6 and CEA showed an AUC of 0.767 for AIS patients, with 57.1% sensitivity at 91.4% specificity in the validation set.

**Conclusions**. IL-6 shows potential as a prospective serum biomarker for the diagnosis of AIS, and the combination of serum IL-6 with CEA may contribute to increased accuracy in AIS diagnosis. However, it is worth noting that further research is still necessary to validate and optimize the diagnostic efficacy of these biomarkers and to address potential sensitivity limitations.

# INTRODUCTION

Lung cancer has become the primary cause of cancer-related deaths worldwide. In China, the incidence and mortality rates of lung cancer have reportedly increased by 224.0% and 195.4% from 1990 to 2019, respectively, indicating a significantly elevated burden of the disease (*Fang et al., 2023*). Lung adenocarcinoma (LUAD) has emerged as a predominant histologic subtype of lung cancer, representing 40–60% of all cases in recent years. This increase can be attributed primarily to the decline in small cell lung cancer and lung squamous cell carcinoma, a trend associated with reduced smoking rates (*Zappa & Mousa, 2016*). Unfortunately, LUAD often remains asymptomatic in its early stages, leading to a high percentage of patients presenting with advanced stage LUAD and a dismal 5-year survival rate of less than 20% (*Jantus-Lewintre et al., 2012*). However, patients with stage 0 LUAD who undergo surgical resection have shown an encouraging 5-year survival rate of up to 100% (*Lin et al., 2022*; *Hwang et al., 2020*). Hence, early diagnosis and timely intervention play vital roles in improving LUAD prognosis.

Currently, low-dose spiral CT is commonly employed for the early screening of LUAD due to its relatively high sensitivity for detecting pulmonary nodules. However, this approach cannot effectively differentiate between lung adenocarcinoma *in situ* (AIS) and benign pulmonary nodules (BPN), as most pulmonary nodules smaller than one cm are frequently associated with benign conditions such as inflammation, fibrosis, or hemorrhage, rather than AIS (*Park et al., 2007*; *Henschke et al., 2012*). Consequently, false-positive results and over-diagnosis have become significant concerns with low-dose spiral CT screening (*Shieh & Bohnenkamp, 2017*). By contrast, serum biomarkers offer a non-invasive approach with the advantages of convenience and safety in screening asymptomatic populations. Unfortunately, there are limitations to the detection sensitivity and specificity of current serum tumor markers, restricting their application value in the early diagnosis of LUAD, particularly in detecting AIS (*Fujita et al., 2004*; *Ma et al., 2011*; *Henry & Hayes, 2012*). Therefore, there is a pressing need to develop new strategies for AIS diagnosis.

Interleukin 6 (IL-6), a key cytokine with diverse functions, has demonstrated potential in regulating cell proliferation and apoptosis by upregulating the expression of specific transcription factors through the IL-6/JAK/STAT3 signaling pathway (*Sansone & Bromberg, 2012*). Notably, recent studies have implicated IL-6 in the occurrence and progression of various malignancies (*Florescu et al., 2023*; *Berger et al., 2023*; *Zhu et al., 2014*). For instance, IL-6 induces associated cancer cell proliferation, apoptosis inhibition, angiogenesis and enhanced drug resistance (*Shintani et al., 2016*; *Wei et al., 2003*). Additionally, aberrantly high levels of serum IL-6 are detected in patients with lung cancer compared to those in benign pulmonary lesions (*Yanagawa et al., 1995*). The ability of serum IL-6 combined with IL-8 and CEA to differentiate lung cancer from health controls (*Yan et al., 2022*) highlights the diagnostic potential of serum IL-6 for lung cancer. As a result, it is worth

exploring whether serum IL-6 detection is conducive to the diagnosis of early-stage LUAD, particularly AIS.

Our pilot study involving a small cohort of 90 cases comprising of 30 AIS patients, 30 BPN patients, and 30 healthy controls (HC), utilizing a chemiluminescence immunoassay, demonstrated that serum IL-6 detection can significantly distinguish the AIS group from the BPN and HC groups. This distinction was evident in the significantly higher levels of serum IL-6 observed in the AIS group compared to those in the BPN and HC groups ($p < 0.05$). The area under the curve (AUC) for serum IL-6 in AIS patients was 0.706, with a sensitivity of 50.0% and specificity of 95.0% (Fig. S1). Therefore, in this study we aim to comprehensively assess the diagnostic value of individual and combined detection of serum IL-6 with traditional tumor markers, such as carcinoembryonic antigen (CEA) and cytokeratin 19 fragment (CYFRA21-1), in AIS patients in a large cohort of 300 cases. This cohort included a training set of 65 AIS, 65 BPN, and 65 HC individuals to develop a predictive model for AIS diagnosis, as well as an independent validation set of 35 AIS, 35 BPN, and 35 HC individuals to validate the prediction model.

## MATERIALS & METHODS

### Study subjects

A total of 100 patients with AIS, who had undergone surgical resection and received a confirmed pathological diagnosis, were recruited, alongside 100 patients with BPN, diagnosed through imaging, and 100 individuals designated as healthy controls (HC), from January 2020 to October 2021 at Fujian Provincial Hospital. Among the subjects, the 100 HC participants received health examinations from the physical examination center and showed no evidence of any disease, including malignancies or BPN. The 100 AIS and 100 BPN patients were recruited following a clinical assessment confirming they did not suffer from cardiovascular disease, autoimmune diseases, pneumonia or other inflammatory diseases, and were not receiving pharmacological treatments including anti-inflammatory and immunotherapy treatments. Before surgery, 5 ml of peripheral blood was collected from each subject, and the serum was separated by centrifugation at 3,000 rpm for 5 min and stored at $-80\,°C$ until use. This study was approved by the Institutional Review Board of Fujian Provincial Hospital, and all participants provided written informed consent (approcal number K2020-01-035). The participants' clinical data from the training set and validation set are shown in Tables 1 and 2, respectively.

### Detection of serum biomarkers
#### Detection of serum IL-6

Serum IL-6 levels were measured using the chemiluminescence immunoassay kit from the C2000 analyzer (Beijing Hotgen, Beijing, China) following the manufacturer's instructions. The cut-off value for serum IL-6 in AIS patients was determined by the receiver operating characteristic (ROC) curve based on the highest discriminant ability (maximum sum of sensitivity and specificity), using the BPN and HC groups as the control.

**Table 1** Clinical data and serum levels of IL-6, CEA, CYFRA21-1 in different groups of the training set.

| Variable | AIS ($n = 65$) | BPN ($n = 65$) | HC ($n = 65$) | $p$-value |
|---|---|---|---|---|
| Age (Year) | 52.8 ± 12.5 | 52.3 ± 13.4 | 50.5 ± 10.7 | >0.05 |
| Gender (%) | | | | >0.05 |
| -Male | 35 (53.8) | 32 (49.2) | 29 (44.6) | |
| -Female | 30 (46.2) | 33 (50.8) | 36 (55.4) | |
| serum level [M(P25-P75)] | | | | |
| IL-6 (pg/ml) | 1.18 (0.83–2.02) | 1.11 (0.81–1.56) | 0.96 (0.63–1.39) | 0.039 |
| CEA (ng/ml) | 1.88 (0.74–4.92) | 1.30 (0.85–2.82) | 1.09 (0.50–1.96) | 0.002 |
| CYFRA21-1 (ng/ml) | 2.20 (1.40–3.42) | 1.48 (1.20–2.71) | 1.94 (1.43–2.96) | 0.054 |

Notes.

AIS, adenocarcinoma *in situ* of lung; BPN, benign pulmonary nodule; HC, healthy control; *p*-value, probability value; M(P25-P75), median(25-75 percent).

**Table 2** Clinical data and serum levels of IL-6, CEA, CYFRA21-1 in different groups of the validation set.

| Variable | AIS ($n = 35$) | BPN ($n = 35$) | HC ($n = 35$) | $p$-value |
|---|---|---|---|---|
| Age (Year) | 55.6 ± 9.1 | 56.3 ± 19.4 | 54.5 ± 15.7 | >0.05 |
| Gender (%) | | | | >0.05 |
| -Male | 15 (42.9) | 17 (48.6) | 19 (54.3) | |
| -Female | 20 (57.1) | 18 (51.4) | 16 (45.7) | |
| serum level [M(P25-P75)] | | | | |
| IL-6 (pg/ml) | 2.08 (0.78-4.98) | 0.94 (0.32–1.56) | 0.76 (0.52–1.76) | 0.002 |
| CEA (ng/ml) | 1.76 (0.83–4.38) | 0.93 (0.78–2.12) | 0.94 (0.50–1.93) | 0.048 |

Notes.

AIS, adenocarcinoma *in situ* of lung; BPN, benign pulmonary nodule; HC, healthy control; *p*-value, probability value; M(P25-P75), median(25-75 percent).

### Detection of serum traditional tumor markers CEA and CYFRA21-1

The levels of serum CEA and CYFRA21-1 were measured using the electrochemilumines-cence immunoassay kit from cobas 602 analyzers (Roche, Mannheim, Germany) following the manufacturer's instructions. The cut-off values for serum CEA and CYFRA21-1 levels in AIS patients were determined by ROC curves based on the highest discriminant ability (maximum sum of sensitivity and specificity) using the BPN and HC groups as the control.

## Statistical analysis

The nonparametric Mann-Whitney U test was used to compare quantitative data between the two groups. For comparisons among more than three groups, the nonparametric Kruskal-Wallis test and one-way ANOVA on ranks (Kruskal-Wallis) test were used. ROC curves were generated using GraphPad Prism 8.0 software, and the area under the ROC curve (AUC) values were calculated. A $p$-value of less than 0.05 was considered statistically significant.

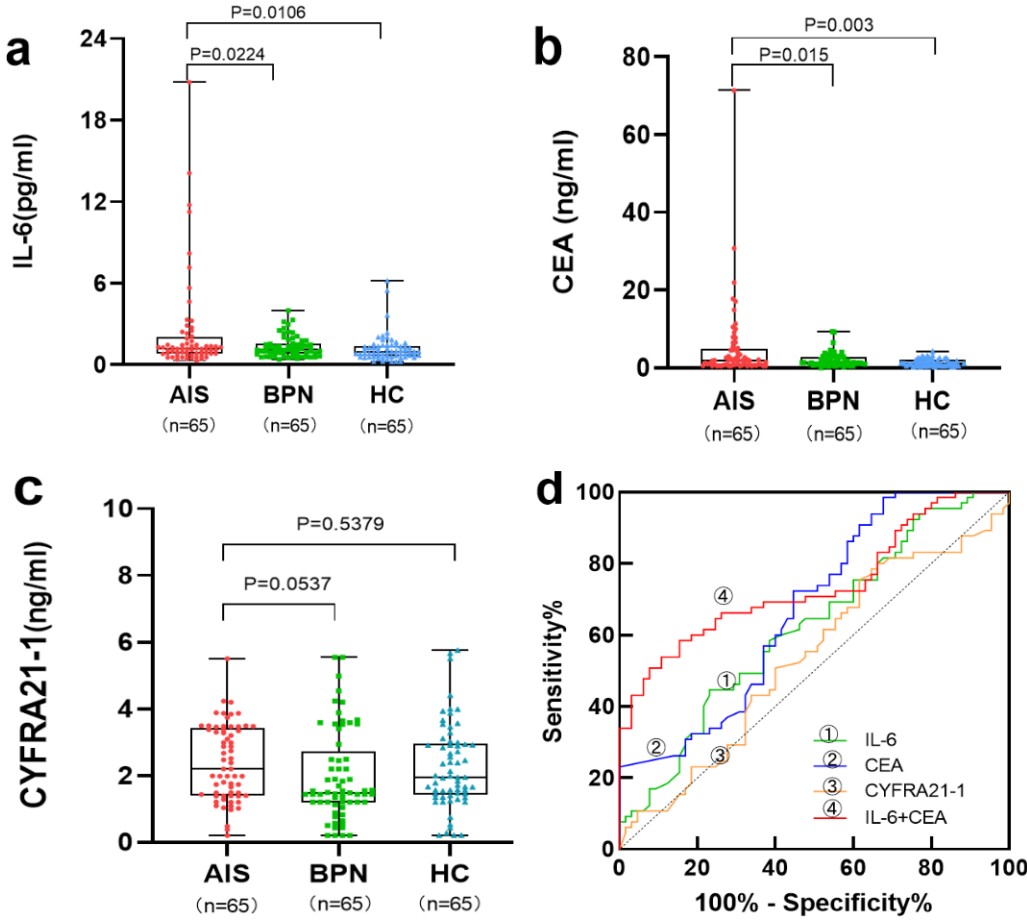

**Figure 1** **Performances of serum IL-6, CEA, and CYFRA21-1 for AIS patients in the training set.**
Boxplot, scatter of serum IL-6 (A), CEA (B), CYFRA21-1 (C), and the ROC curves of serum IL-6, CEA,
CYFRA21-1, and IL-6+CEA (D).

## RESULTS

### The levels of serum biomarkers in the training set

As shown in Table 1 and Figs. 1A–1C, the levels of serum IL-6 and CEA were significantly
higher in the AIS group than in the BPN/HC group ($p < 0.05$). However, there was no
significant difference in serum CYFRA21-1 levels between the AIS group and the BPN ($p$
$=0.537$)/HC ($p = 0.537$) group.

### Levels of serum biomarkers in the validation set

Table 2 and Figs. 2A, 2B show the levels of serum IL-6 and CEA in the AIS group, which
were significantly higher than those in the BPN/HC group ($p < 0.05$).

### Diagnostic performance of individual and combined detection of serum biomarkers in the training set

Table 3 and Fig. 1D present the results of individual and combined detection of serum
biomarkers for AIS patients in the training set. Serum IL-6 showed an AUC of 0.622 for AIS

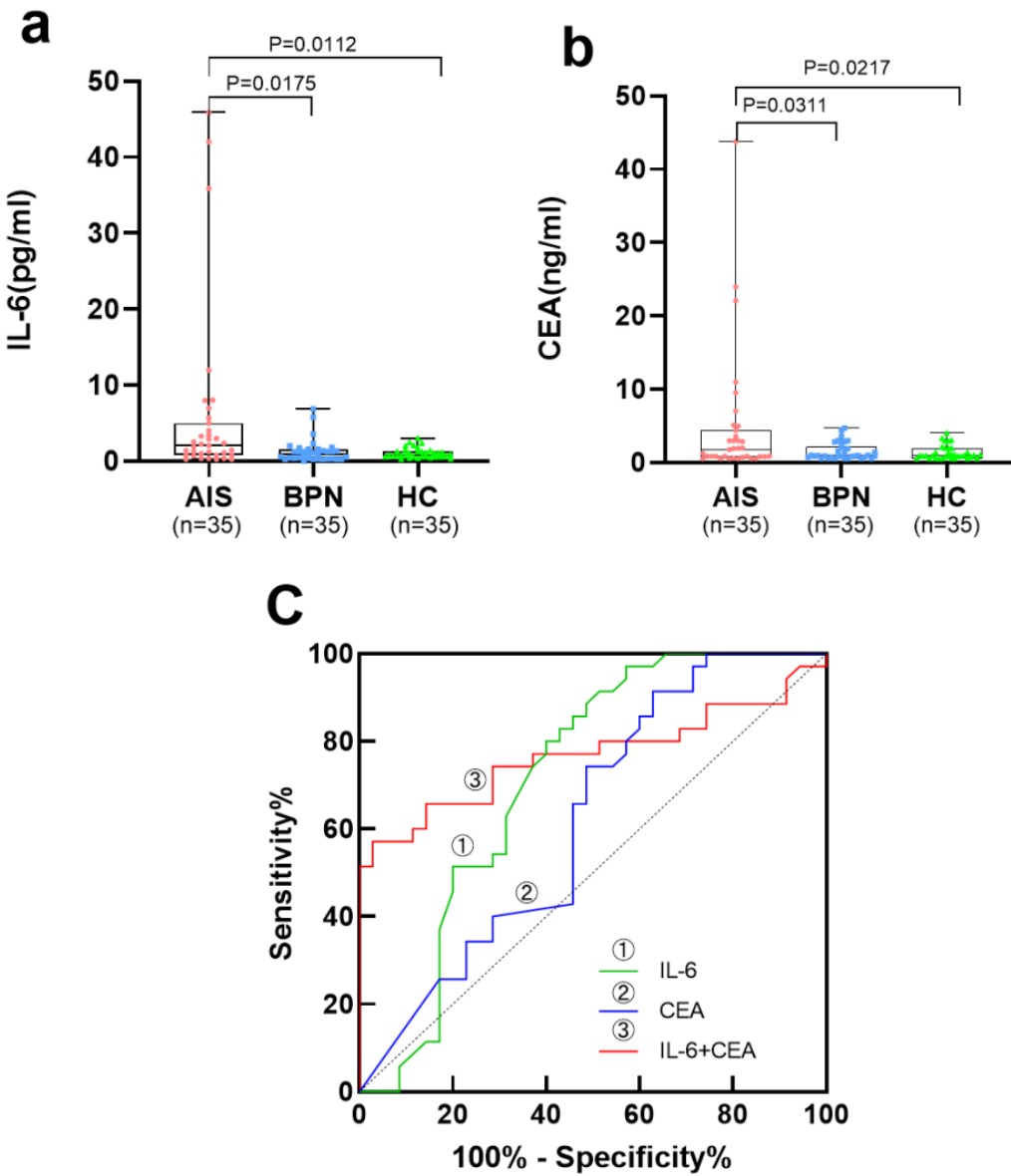

**Figure 2** **Performances of serum IL-6 and CEA for AIS patients in the validation set.** Boxplot, scatter of serum IL-6 (A), CEA (B), and the ROC curves of serum IL-6, CEA, and IL-6+CEA (C).

patients, with a sensitivity of 23.1% at a specificity of 90.7%. Serum CEA and CYFRA21-1 exhibited an AUC of 0.672 for AIS patients, with a sensitivity of 24.6% at a specificity of 97.6%, and an AUC of 0.538 for AIS patients, with a sensitivity of 30.7% at a specificity of 80.0%, respectively. Among all possible combinations, the panel of serum IL-6 and CEA demonstrated the best discriminative ability for AIS, with the highest AUC of 0.739 (95% CI [0.653–0.826]), a sensitivity of 60.0% at a specificity of 95.4%, and the highest positive predictive value (PPV) of 86.7% and negative predictive value (NPV) of 82.7%.
**Table 3  Comparison of the diagnostic performances of serum IL-6, CEA, and CYFRA21-1 for AIS patients in the training set.**

| Marker | AUC | SE | 95% CI | Sensitivity (%) | Specificity (%) | PPV (%) | NPV (%) | *p*-value |
|--------|-----|-----|--------|-----------------|-----------------|---------|---------|-----------|
| IL-6 | 0.622 | 0.049 | 0.526–0.718 | 23.1 | 90.7 | 55.6 | 70.2 | 0.017 |
| CEA | 0.672 | 0.047 | 0.579–0.764 | 24.6 | 97.6 | 84.2 | 72.2 | 0.001 |
| CYFRA21-1 | 0.538 | 0.051 | 0.438–0.638 | 30.7 | 80.0 | 43.4 | 69.8 | 0.456 |
| IL-6+CEA | 0.739 | 0.044 | 0.653–0.826 | 60.0 | 95.4 | 86.7 | 82.7 | <0.0001 |
| IL-6+CYFRA21-1 | 0.635 | 0.049 | 0.530–0.740 | 36.9 | 96.2 | 82.8 | 75.3 | 0.008 |
| CEA+CYFRA21-1 | 0.664 | 0.043 | 0.579–0.749 | 70.8 | 59.2 | 75.0 | 72.5 | <0.0001 |
| IL-6+CEA+CYFRA21-1 | 0.721 | 0.045 | 0.633–0.809 | 43.1 | 95.4 | 82.4 | 77.0 | <0.0001 |

Notes.
AUC, area under the curve; SE, standard error; 95% CI, 95% confidence interval; PPV, positive predictive value; NPV, negative predictive value; *p*-value, probability value.

**Table 4  Comparison of the diagnostic performances of serum IL-6, CEA, andCYFRA21-1 for AIS patients in the validation set.**

| Marker | AUC | SE | 95% CI | Sensitivity (%) | Specificity (%) | PPV (%) | NPV (%) | *p*-value |
|--------|-----|-----|--------|-----------------|-----------------|---------|---------|-----------|
| IL-6 | 0.716 | 0.064 | 0.590–0.843 | 45.7 | 90.0 | 66.7 | 76.8 | 0.002 |
| CEA | 0.621 | 0.068 | 0.488–0.755 | 20.0 | 100 | 100 | 71.4 | 0.081 |
| IL-6+CEA | 0.767 | 0.060 | 0.650–0.886 | 57.1 | 91.4 | 76.9 | 81.0 | <0.0001 |

Notes.
AUC, area under the curve; SE, standard error; 95% CI, 95% confidence interval; PPV, positive predictive value; NPV, negative predictive value; *p*-value, probability value.

### The diagnostic performances of individual and combined detection of serum IL-6 and CEA for AIS in the validation set

In the validation set, serum IL-6 presented an AUC of 0.716 for AIS patients, with a sensitivity of 45.7% at a specificity of 90.0%. Serum CEA presented an AUC of 0.621 for AIS patients, with a sensitivity of 20.0% at a specificity of 100.0%. The combination of serum IL-6 and CEA presented an AUC of 0.767, with a sensitivity of 57.1% at a specificity of 91.4% (Table 4, Fig. 2C).

## DISCUSSION

LUAD cells exhibit highly malignant behaviors, leading to rapid metastasis and a poor prognosis. The early detection of AIS, a preliminary stage of LUAD, is challenging due to the lack of efficient diagnostic approaches (*Naito et al., 2016*). Effective discrimination and timely treatment of AIS is crucial for patients with LUAD. IL-6, a key cytokine in the complex cytokine network, has been found to play a vital role in the occurrence and development of various malignancies, in addition to its traditional role in immune regulation (*Hirano, 2021*). Numerous studies have shown that IL-6 is overexpressed not only in cancerous tissues but also in the sera of patients with different malignancies, such as lung cancer, multiple myeloma, leukemia, gastric cancer, and colon cancer. High IL-6 levels have been closely associated with cancer progression and poor prognosis for patients (*Zhu et al., 2014*; *Shintani et al., 2016*; *Gu et al., 2021*; *Bataille et al., 1989*; *Kasuga et al., 2001*).

In our pilot study, we observed the ability of serum IL-6 to differentiate AIS patients from BPN and HC patients. This prompted us to explore the potential of serum IL-6 as a

diagnostic biomarker for AIS patients.In a subsequent analysis of 300 retrospective serum samples including 100 AIS, 100 BPN, and 100 HC samples, we obtained the expected results. We found that levels of serum IL-6 and the traditional tumor marker CEA were significantly higher in the AIS group compared to those in the BPN/HC group ($p < 0.05$). However, there were no significant differences in serum CYFRA21-1 levels between the AIS group and BPN ($p = 0.0537$)/HC ($p = 0.537$) group. These findings indicate that IL-6 could serve as a potential serum biomarker for the diagnosis of AIS.

Our findings from the training set allowed us to clearly define the diagnostic performance of serum IL-6 for AIS patients, presenting an AUC of 0.622, with a sensitivity of 23.1% and specificity of 90.7%. We then successfully validated the diagnostic value of serum IL-6 for AIS patients in an independent validation set.

Studies suggest that CEA may lack sensitivity, especially in early-stage lung cancer. It is often associated with advanced stages or metastasis rather than early detection (*Li et al., 2021*). While CYFRA 21-1 is useful, its sensitivity may be limited, particularly in distinguishing between benign and malignant lesions.Factors such as inflammatory conditions and certain benign lung diseases may contribute to false positives, affecting its diagnostic accuracy (*Moro et al., 1995*). Considering the limited sensitivity of individual detection, we assessed the diagnostic performance of IL-6 combined with the traditional tumor markers CEA and CYFRA21-1 in AIS patients. Among all possible combinations, a panel of serum IL-6 with CEA presented an optimal diagnostic performance for AIS, supported by an increased AUC of 0.739 and a sensitivity of 60.0% at a high specificity of 95.4%. This combination also exhibited diagnostic potential for AIS in an independent vh an AUC of up to 0.767, a sensitivity of 57.1%, and specificity of 91.4%.

To the best of our knowledge, this study represents the first attempt to devise a screening strategy that effectively distinguishes AIS from BPN, utilizing a serological panel with commendable sensitivity while maintaining a high specificity. Nevertheless, we have to acknowledge that the diagnostic value of this serological panel for AIS should be further verified by multi-center research in the future. Multi-center studies would comprehensively assess patients from different populations, regions, and disease stages, aiding in verifying the universality and generalizability of our findings. This research design can provide stronger evidence, supporting the feasibility of our serological panel in diverse clinical contexts and offering more practical guidance for clinical practice. Further studies should also be conducted to uncover additional serological biomarkers that will enhance the sensitivity of AIS diagnosis, ultimately improving the 5-year survival rate and prognosis of LUAD patients.

## CONCLUSIONS

In summary, our study highlights the potential of serum IL-6 as a promising biomarker for early-stage LUAD, specifically AIS. The combined use of IL-6 with the traditional tumor marker CEA showed significant diagnostic performance in our analysis of 300 serum samples.While our findings are promising, validation across larger and more diverse samples is necessary. The complexity of AIS diagnosis in early-stage LUAD requires further

research to confirm the applicability of IL-6 across different populations. Future studies should explore additional serum biomarkers to enhance AIS diagnostic sensitivity, and multi-center research is crucial for validating and applying these biomarkers in various clinical settings. In essence, our study provides a hopeful tool for early LUAD screening, emphasizing the need for ongoing research to refine and confirm the diagnostic value of these potential biomarkers, aiming to improve patient prognosis and survival rates.

### Funding
This study was supported by Medical Vertical Project of Fujian Province (Grant No. 2020CXB001), the Joint fund of Science and Technology Innovation of Fujian province (Grant No. 2021Y9024), and the Key Project of Natural Science Foundation of Fujian province (Grant No. 2022J02048). The funders had no role in study design, data collection and analysis, decision to publish, or preparation of the manuscript.

### Grant Disclosures
The following grant information was disclosed by the authors:
Medical Vertical Project of Fujian Province: 2020CXB001.
Science and Technology Innovation of Fujian province: 2021Y9024.
Key Project of Natural Science Foundation of Fujian province: 2022J02048.

### Competing Interests
The authors declare there are no competing interests.

### Author Contributions
- Jing Pan conceived and designed the experiments, performed the experiments, analyzed the data, prepared figures and/or tables, authored or reviewed drafts of the article, and approved the final draft.
- Wanzhen Zhuang performed the experiments, prepared figures and/or tables, and approved the final draft.
- Yu Xia performed the experiments, prepared figures and/or tables, and approved the final draft.
- Zhixin Huang performed the experiments, prepared figures and/or tables, and approved the final draft.
- Yue Zheng analyzed the data, prepared figures and/or tables, and approved the final draft.
- Xin Wang analyzed the data, authored or reviewed drafts of the article, and approved the final draft.
- Yi Huang conceived and designed the experiments, authored or reviewed drafts of the article, and approved the final draft.

### Human Ethics
The following information was supplied relating to ethical approvals (i.e., approving body and any reference numbers):

Ethics Committee of Fujian Provincial Hospital

## Data Availability

The raw measurements are available in the Supplementary Files.

## Supplemental Information

Supplemental information for this article can be found online at http://dx.doi.org/10.7717/peerj.17141#supplemental-information.

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
