# Peer review of "Combined detection of serum IL-6 and CEA contributes to the diagnosis of lung adenocarcinoma in situ"

_PeerJ, doi:10.7717/peerj.17141_

## Round 0.1 · original submission · Major Revisions

The paper needs major revision before it can be considered for publication. Please address all the issues raised by the referees.

Reviewer 1 ·

Basic reporting

The paper was well written and easy to understand basically. However, some literatures were not well cited, for example, ref. 13 did not related to IL-6; ref. 14 was not conduced in cancer patients; ref. 16 was just a meta-analysis for the role of IL-6 in prognosis in cancer patients; ref. 22 was a in vitro study; ref. 23 was unrelated to the prognosis. The background was clear, and the structures, figures and tables are also clear.

Experimental design

The design of the study was reasonable, and the question was well defined. The investigation was rigorous but with some questions, which mainly concerning the sample size. Since the authors have a previous investigation for their assumption, the enlarged sample size for present study should be rigorous calculated based on previous data. And why they think 300 was enough for present study?

Validity of the findings

The results of this study were novelty and may be helpful in clinic to determine the AIS from benign pulmonary nodes and others. All the data were well analyzed, and the statistical methods were well described and right. The conclusion for this study was basically right, however, it was very important to conduct a comparison for the AUCs of IL-6, CEA and IL6+CEA in this study (the method of DeLong et al.) since the conclusion could be confirmed only when the statistical difference was significant by the analysis.

Additional comments

1. The discussion of the study was very poor; it just repeated the results of the study.
2. The exact statistical results in the paper should be listed including all the Tables rather than >0.05.
3. Some sentences in the paper are hard to understand, for example: line 52-54.

Reviewer 2 ·

Basic reporting

The manuscript by Pan et Al. explains a plain yet well designed study, in which objectives and methods are clear.
The results are limited in terms of practical use, but they can generate additional research questions, although they cannot help generate a diagnostic algorythm right now.

In general, the manuscript seems a bit "too simple" and can benefit from some integrations, as reported in the subsequent sections, in order to provide useful information to the readers. Hence, I recommend some major revisions before considering this work for publication.

Experimental design

The design of this research is quite clear; however, some explanations and additions are suggested:

1) You should explain in more detail whether patients with BPN and AIS underwent surgical resection. This non-negligible information is pivotal for the study at it provides the histologic diagnosis of each nodule. However, the information of surgery is only implied (when you report, in line 103, that the patients underwent blood collection before surgery) in the methods, and it should be explained with more detail.

2) While I get the point of comparing health controls and patients with benign nodules as a single control group, I am also curious to understand whether comparing only AIS and BPN can provide the same results. This could be an exploratory analysis to enrich the manuscript, based on the fact that in clinical practice we need to discriminate between AIS and BPN as the subject of our investigations. An exploratory analysis on these sub-sets can add something to this work and provide more information, in addition to what is already written.

3) The clinical characteristics reported in the main tables appear to be limited to gender and age, and I did not understand whether multivariate analyses involving these characteristics with the biomarkers have been performed. Additionally, I believe additional factors, such as smoking habit, size of the nodule (or even number of nodules) and shape (i.e. spiculated) can be relevant if we want to develop a diagnostic tool for AIS vs. BPN. Have you got these information? Can you provide a multivariate analysis in order to develop a signature with both clinical and circulating parameters? This would be helpful.

Validity of the findings

The findings are interesting, albeit limited. Please consider the following observations:

1) Since the definition of AIC (adenocarcinoma in situ) is pivotal for understanding the meaning of this manuscript, you should provide a more extensive focus on the lung adenocarcinoma in situ compared to small invasive adenocarcinoma. Did you exclude patients with small (T1) invasive adenocarcinoma to include only AIS? Then when did you make patient selection? And what about early stage squamous cell lung cancer? You should explain the limits of this study for these malignancies, as it can be challenging to discriminate AIS from small, early stage LUAD and SCC. Additionally, a more precise definition of AIS would be helpful.

2) Some attempt to explain the poor sensitivity of individual biomarkers should be made in the discussion section. Additionally, the authors should try to provide some literature-based explanation of the significant improvement in terms of sensitivity combining all the biomarkers. Similarly, the authors should describe in more detail the role of IL-6, since CEA and CYFRA 21-1 are more acknowledged in lung cancer (by contrast, IL-6 has become popular with other, non-cancer-related subjects, like infective diseases).

3) Finally, the authors should acknowledge that the impact of their finding is still limited, as these results are not sufficient to promote a screening before surgery (I find hard to develop a management algorythm based on these observations). In this context, while the authors correctely encouraged further, multi-centric studies, this concept should be stressed more.

Additional comments

1) Some language editing is required throughout the whole manuscript. For istance, in the abstract, you report "with 60.0% sensitivity at 95.4% specificity". The word "at" should be replaced with "and".

2) Time consecution. In some parts of the manuscript you use future tense (i.e. line 91), while in other cases you use past tense (i.e. line 97). Please re-phrase your manuscript consistently.

3) The pilot study with 90 samples is reported multiple times in the manuscript, both in the introduction and in the discussion. Please remove the redundant reference from the discussion.

---

## Round 0.2 · Major Revisions

I appreciate that the authors have answered all issues raised by the reviewers. Yet still there are several statements in the abstract, in the discussion and in the conclusions that overstate the potential of IL6 and its combination with CEA as biomarkers. There is a problem with sensitivity and this needs to be acknowledged. Please comment a little less enthusiastically on your findings and more clearly indicate that more research is needed to identify really robust prognostic procedures.

---

## Round 0.3 · accepted · Accept

All issues raised have been adequately addressed.